# Abnormal Brain-Derived Neurotrophic Factor Exon IX Promoter Methylation, Protein, and mRNA Levels in Patients with Major Depressive Disorder

**DOI:** 10.3390/jcm8050568

**Published:** 2019-04-26

**Authors:** Men-Ting Hsieh, Chin-Chuen Lin, Chien-Te Lee, Tiao-Lai Huang

**Affiliations:** 1Department of Psychiatry, Kaohsiung Chang Gung Memorial Hospital and Chang Gung University College of Medicine, Kaohsiung 83301, Taiwan; angel0073536@hotmail.com (M.-T.H.); hijks@adm.cgmh.org.tw (C.-C.L.); 2Division of Nephrology, Department of Internal Medicine, Kaohsiung Chang Gung Memorial Hospital and Chang Gung University College of Medicine, Kaohsiung 83301, Taiwan; ctlee33@cgmh.org.tw

**Keywords:** brain-derived neurotrophic factor, DNA methylation, major depressive disorder

## Abstract

Brain-derived neurotrophic factor (BDNF) exon IX promoter methylation levels, serum BDNF protein levels, and serum mRNA levels were investigated in patients with major depressive disorder (MDD) and healthy controls. Over two years, 51 patients with MDD and 62 healthy controls were recruited. Peripheral blood was drawn from all participants to analyze the BDNF exon IX promoter methylation levels as well as serum BDNF protein and mRNA levels, at baseline and after four weeks of antidepressant treatment. Methylation sequential analysis showed that patients with MDD (*n* = 39) had a higher methylation level at CpG site 217 and lower methylation levels at CpG site 327 and CpG site 362. Drug responders (*n* = 25) had a higher methylation level at CpG site 24 and CpG site 324 than the non-responders (*n* = 11). Patients with MDD had a lower serum BDNF protein and mRNA levels than the healthy controls. In conclusion, these results showed that BDNF exon IX promoter methylation levels, serum BDNF protein level, and serum BDNF mRNA level could contribute to the pathophysiology of a major depressive disorder.

## 1. Introduction

Brain-derived neurotrophic factor (BDNF) has been established as a candidate molecule for the pathophysiology of neuropsychiatric disorders, such as schizophrenia [1] and major depressive disorder (MDD) [2,3,4,5]. Many past studies have shown the relationships between decreased serum BDNF protein levels and MDD [5,6,7,8,9,10,11,12]. Antidepressant treatment has also been shown to increase BDNF levels [13]. The relationships between BDNF polymorphisms and MDD have also been extensively studied [5,14,15,16,17,18,19,20,21,22]. Generally speaking, lower serum BDNF protein levels were frequently observed in the patients with MDD, as compared with healthy controls, and the presence of BDNF polymorphisms had been associated with lower BDNF protein levels and certain clinical phenotypes, such as suicide.

However, investigations on candidate molecules and their pathways have not fully explained the pathophysiology of various neuropsychiatric disorders. In the recent years, epigenetic investigations of various neuropsychiatric disorders have garnered more attention. Epigenetic modifications such as DNA methylation, histone modifications, and non-coding RNAs can affect gene activity and expression, without alterations in the DNA sequence. Epigenetic regulation in the postmortem brain tissues and peripheral blood cells of psychiatric patients has been studied [5,14,23,24,25,26,27,28]. Some of those works focused on epigenetic alterations of promoters of BDNF gene and its receptor tyrosine kinase B (TrkB) [5,9,14,24,25,28].

In the animal models of depression, the epigenetic modifications of the BDNF promoters have been investigated, particularly on BDNF exons I, IV, and IX [29,30,31,32]. In clinical samples, studies had also investigated the relationship between BDNF and TrkB methylation status and antidepressant response, as well as their association with suicide [14,19,25,33]. In past studies, methylation levels of BDNF exon I and IV promoters in peripheral blood were studied more extensively, and were found be to be associated with severe mental disorders, such as schizophrenia [34,35], bipolar mania [36] and MDD [5,24,28,37]. Methylation level of CpG site 87 of BDNF exon IV promoters in patients with MDD could be used to predict antidepressant response [28]. However, data on the CpG sites near rs6265 (equivalent to the G196A or Val66Met polymorphism) in the BDNF region in peripheral blood has not been studied before, especially at BDNF exon IX (coding exon) [30,38].

In this study, we aimed to investigate the associations of BDNF promoter exon IX DNA methylation, BDNF protein, and mRNA levels in the peripheral blood of patients with MDD and healthy controls.

## 2. Experimental Section

### 2.1. Participants

From August 2010 to July 2012, patients with MDD and healthy controls were evaluated according to DSM-IV criteria using a semi-structured interview or the Chinese Health Questionnaire-12 [39]. Institutional review board approval was obtained from the ethics committee of our medical center, a tertiary hospital in southern Taiwan. The assessments were done by the same senior psychiatrist. The 17-item Hamilton Depression Rating Scale (HAM-D) was used to assess severity of MDD [40]. The minimum baseline score of the 17-item HAM-D was 18. Antidepressant treatment responders were defined as those with a minimum 50% decrease in the HAM-D total score after 4 weeks of medical treatment. The antidepressants used were all selective serotonin reuptake inhibitors (SSRI), including fluoxetine 20–40 mg/day, paroxetine 20–40 mg/day, and escitalopram 20–40 mg/day. Patients were allowed the combined use of benzodiazepines (i.e., lorazepam 3 mg/day) or hypnotics (i.e., zolpidem 10 mg/day). No patient received mood stabilizers or antipsychotic drugs. The patients did not have systemic diseases, including cardiovascular, liver and thyroid diseases. All patients did not receive any drugs for at least 2 weeks before entering the study. They were also not heavy smokers or alcohol-dependent. The patients had the ability to complete the written informed consent.

### 2.2. Assessments of Serum BDNF Protein, mRNA, and DNA Methylation Levels

Venous blood (15 mL) was obtained from each sample to analyze the BDNF protein, mRNA, and exon IX promoter methylation levels in all participants at baseline. Those biological markers in the patients with MDD were checked again after four weeks of antidepressant treatment.

#### 2.2.1. DNA Isolation and Bisulfite Treatment

Genomic DNA was isolated using the Easy Blood genomic DNA purification kit (Genemark, Taichung, Taiwan), following the manufacturer’s instructions. DNA from the Lymphocytes. Genomic DNA (500 ng) was converted with sodium bisulfite using the EZ DNA methylation kit (Zymo Research, Irvine, CA, USA) [41,42]. The concentration of sodium bisulfate-treated DNA was measured using an ND-1000 spectrophotometer (NanoDrop Technologies, Wilmington, DE, USA), and 20 ng of treated DNA was used in a region-specific PCR.

#### 2.2.2. Pyrosequencing Analysis

We chose 14 CpG sites near rs6265 in the BDNF exon IX. Pyrosequencing analysis was performed after polymerase chain reaction (PCR). The detailed information of those sites was the same as previously described [43].

#### 2.2.3. Serum BDNF Protein Levels

Serum BDNF protein levels were measured with a commercially available ELISA kit, as previously described [43].

#### 2.2.4. mRNA Isolation and Reverse Transcription-Polymerase Chain Reaction

RNA was isolated after peripheral blood was collected. The RNA was then reverse-transcribed into cDNA, and quantitative PCR was performed. The detailed information of those sites was the same as previously described [43].

### 2.3. Data Analysis

All results are expressed as mean ± standard deviation (SD). Statistical differences in serum BDNF protein level, mRNA, and exon IX CpG sites methylation density levels of patients with MDD and healthy controls were determined by analysis of variance (ANOVA), analysis of co-variance (ANCOVA) with age or body mass index (BMI) adjustment, or independent *t*-test. Comparison of data acquired at baseline and after four weeks of antidepressant treatment was made with paired *t*-test. *p* values < 0.05 were considered to be statistically significant.

## 3. Results

### 3.1. Demographic Data

From August 2010 to July 2012, 51 patients with MDD and 62 healthy controls were recruited for this study. Their demographic data and serum BDNF mRNA levels at baseline of all participants are shown in Table 1.

### 3.2. BDNF Exon IX Promoter Methylation Sequential Analysis

Of the 113 participants, 39 patients with MDD (mean age = 42.4 ± 10.8 years, BMI = 22.1 ± 4.1 kg/m^2^, male/female = 9/30) and 62 healthy controls completed the BDNF exon IX promoter methylation sequential analysis. The methylation density levels of the 14 CpG sites evaluated are shown in Figure 1.

Using independent *t*-test, patients with major depressive disorder had a higher methylation level at CpG site 217 (*t* = 4.427, *p* = 0.000) and lower methylation levels at CpG site 327 (*t* = 2.857, *p* = 0.016) and CpG site 362 (*t* = 2.887, *p* = 0.017) than the healthy controls. Using ANCOVA with age adjustment, no statistically significant differences were found in methylation levels of the 14 CpG sites. Using ANCOVA with BMI adjustment, significant differences were detected at CpG site 362 (*F* = 4.080, *p* = 0.046) and CpG site 391 (*F* = 5.072, *p* = 0.027). Using ANCOVA with both age and BMI adjustment, a significant difference was found at CpG site 348 (*F* = 4.003, *p* = 0.035).

Drug responders (*n* = 25) had significantly higher methylation levels at CpG site 24 (*t* = 2.273, *p* = 0.029) and CpG site 324 (*t* = 2.252, *p* = 0.031) than non-responders (*n* = 11), using independent *t*-test (Figure 2). Between patients with (*n* = 12) and without a suicide attempt (*n* = 27), no significant differences were detected in the methylation levels of 14 CpG sites.

### 3.3. Serum BDNF Protein Level

The serum BDNF protein levels of 48 patients with MDD and 62 healthy controls were assessed. The mean levels of serum BDNF protein were 5.6 ± 4.5 ng/mL and 7.9 ± 3.2 ng/mL, respectively, in patients with MDD and the healthy controls (Figure 3). Using an independent *t*-test, patients with MDD had significantly lower BDNF protein levels than healthy controls (*t* = 3.177, *p* = 0.002). No statistically significant difference was detected between patients and controls when analyzed with ANCOVA with age adjustment (*F* = 0.044, *p* = 0.835), ANCOVA with BMI adjustment (*F* = 0.902, *p* = 0.344), or ANCOVA with age and BMI adjustment (*F* = 1.849, *p* = 0.177).

After four weeks of antidepressant treatment, serum BDNF protein levels were 6.3 ± 5.6 ng/mL. Using a paired *t*-test, no significant changes were detected (*t* = 0.658, *p* = 0.512).

### 3.4. Blood BDNF mRNA

The BDNF mRNA levels of 51 patients with MDD and 62 healthy controls were checked. The mean levels of serum BDNF mRNA were 1.6 ± 1.6 and 2.6 ± 1.8, respectively, in patients with MDD and the healthy controls (Figure 4). Using an independent *t*-test, patients with MDD had significantly lower BDNF mRNA levels than the healthy controls (*t* = 3.120, *p* = 0.002). No statistically significant difference was detected between patients and controls when analyzed with ANCOVA with age adjustment (*F* = 2.283, *p* = 0.134), ANCOVA with BMI adjustment (*F* = 0.213, *p* = 0.646), or ANCOVA with age and BMI adjustment (*F* = 0.019, *p* = 0.891).

After four weeks of antidepressant treatment, serum BDNF mRNA levels of 48 patients were measured, with a mean level of 1.8 ± 1.4. Using a paired *t*-test, no significant changes were detected (*t* = 1.586, *p* = 0.120).

## 4. Discussion

The most important findings were that patients with MDD had a significantly higher methylation level at CpG site 217 and lower methylation levels at CpG site 327 and CpG site 362 in BDNF exon IX than the healthy controls, along with lower BDNF protein and mRNA levels. BDNF gene has many exons, but most earlier DNA methylation studies focused on BDNF exon I and IV, and data on exon IX was relatively scarce. In a Japanese study, 29 out of 35 investigated CpG sites in BDNF exon I were differentially methylated between patients with MDD and healthy controls, but no significant difference was detected in methylations of BDNF exon IV [24]. D’Addario et al. reported higher methylation of BDNF exon I promoter and lower BDNF gene expression in patients with MDD [44]. In a study sampling patients with MDD as well as bipolar disorder, patients with MDD had higher BDNF exon I promoter methylation levels than patients with bipolar disorder and healthy controls [45]. Their follow-up study found higher BDNF exon I promoter methylation levels in both patients with MDD and patients with bipolar II disorder, but not in patients with bipolar I disorder; when analysis was performed based on the mood state instead of diagnosis, the study also found that methylation levels of patients in a depressive state were significantly higher, compared to the levels of patients in manic/mixed states [46]. In a study using buccal tissues of geriatric patients with MDD, DNA methylation levels of both BDNF promoters I and IV were associated with depression at baseline, and chronic late-life depression over the 12-year follow-up period [47]. Three single nucleotide polymorphisms, including rs6265, were found to modify the association between depression and BDNF promoter I methylation [47]. Those studies provided a lot of data associating abnormal DNA methylation levels of BDNF exon I, and to some extent, BDNF exon IV, with MDD. In several Korean studies, the DNA methylation of the BDNF promoter region was evaluated using four to seven CpG sites in CpG-rich region of the promoter between −694 and −577. In a follow-up study of patients with post stroke depression, a higher BDNF methylation level was associated with prevalence, persistence, and incidence of depression, and with worsening depressive symptoms over follow-up [48]. In patients with breast cancer after mastectomy, a higher BDNF methylation level was associated with the diagnosis of depression and with more severe symptoms throughout the study period [49]. In geriatric patients with depression, a higher BDNF methylation level was associated with both the prevalence and incidence of depression diagnosis, as well as symptom severity [50]. In patients with acute coronary syndrome evaluated for depressive disorder, a higher BDNF methylation level was associated with prevalent depressive disorder at baseline and with its persistence at follow-up [51]. In patients with MDD, BDNF promoter methylation level was inversely correlated with the integrity of anterior corona radiata [52]. In patients with recurrent MDD, higher levels of BDNF promoter methylation were closely associated with the reduced cortical thickness, and serum BDNF levels were also significantly lower in MDD [53]. In geriatric women, a higher BDNF DNA methylation level was detected in those with anxiety/depression compared to healthy controls [54]. The majority of the aforementioned studies increased methylation levels in patients with MDD [55]. However, there had been few studies focusing on DNA methylation levels of BDNF exon IX promoter.

In terms of BDNF polymorphism rs6265, we did not find any significant difference on CpG site 140 methylation. The findings on rs6265 and DNA methylation levels were not always consistent in the past. Three SNPs, including rs6265, were found to modify the association between late-life depression and BDNF exon I promoter methylation levels [47]. The presence of Met allele of rs6265 was associated with prevalent post stroke depression, but not with persistent and incident types, nor with depression severity. [48]. In geriatric women, a greater BDNF DNA methylation was observed in those with anxiety/depression compared to healthy controls, and interestingly, the difference was more pronounced in patients with the rs6265 CT genotype than with the CC genotype [54]. Other studies noted no significant association with rs6265, however. In a study involving patients with MDD and bipolar disorder, no significant genotype-methylation interactions were found with rs6265 [45]. No significant interaction was detected between the BDNF methylation level and rs6265 in patients with breast cancer after mastectomy [49] or in geriatric patients with depression [50,56]. While BDNF rs6265 was widely investigated in MDD, its significance with epigenetics required further investigations.

Regarding suicide, in our study, no significant differences in methylation levels were found in the 14 CpG sites between patients with and without a suicide attempt. Most earlier studies found an association between increased BDNF methylation levels with suicide. In the postmortem brain samples from suicide subjects, BDNF promoter IV methylation levels were significantly increased, as compared to controls [37]. A higher level of BDNF promoter methylation was significantly associated with suicide history, suicidal ideations, and suicide scale score [5]. A higher BDNF methylation level was also significantly associated with both the prevalence and the incidence of suicidal ideation [56]. Suicide is a severe clinical symptom in MDD, and its association with DNA methylation warrants further research.

In our study, antidepressant responders had higher methylation levels at CpG site 24 and CpG site 324 of BDNF exon IX than non-responders. Earlier studies also showed interactions between BDNF methylation status with treatment response. Patients treated with both mood stabilizers and antidepressants had decreased BDNF exon I promoter methylation compared to patients treated with antidepressants alone [44]. Lack of methylation of CpG site-87 of BDNF exon IV promoter was associated with treatment nonresponse in patients with MDD [28]. However, in the follow-up study of Tadic et al.’s work, the predictor function of CpG-87 of BDNF exon IV promoter was not replicated, but in a subgroup of patients with severe depression, patients with hypermethylation at CpG-87 had significantly higher remission rates than patients without a methylation [57]. In patients after acute coronary syndrome, increased BDNF methylation was associated with that finding that escitalopram was more effective than placebo for treating depressive disorder, and this effects lead to prevent persistent depressive disorder [51]. In addition to medical treatment, electroconvulsive therapy (ECT) could be associated with BDNF DNA methylation. Patients remitting under ECT had significantly lower mean promoter methylation rates of BDNF exon I, IV and VI, especially concerning the exon I promoter, compared to non-remitters [58]. Antidepressant treatment response is an important clinical indicator, and its association with epigenetic biomarker could pave the way for individualized medicine in the future.

Our data simultaneously showed that patients with MDD had lower BDNF protein and mRNA levels than the healthy controls, which was compatible with many previous reports [12,59], but not with other reports [11,60].

In conclusion, these results showed that patients with MDD and healthy controls had significantly different BDNF exon IX promoter methylation levels, BDNF protein level, and mRNA level in peripheral blood. Our data also showed that some CpG sites were differentially methylated between antidepressant responders and non-responders. The mainstream theory is that increased BDNF promoter methylations would lead to impaired DNA expression, thus causing a decrease in protein production. However, different studies tend to utilize different approaches, such as the number of CpG sites surveyed (from 4 to 35 in our citations) and how their methylation levels were calculated (as individual levels vs. a total or global level). Those methodological differences make it difficult to draw a unifying conclusion to develop reliable diagnostic or outcome prediction tools, but as we gain more and more understanding of the interconnecting mechanisms of various pathways, we should in turn develop a better understanding of the underlying pathophysiology of psychiatric disorders.

## Figures and Tables

**Figure 1 jcm-08-00568-f001:**
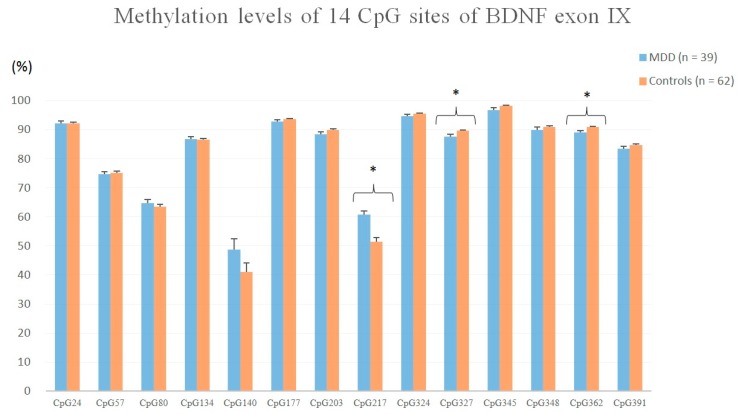
Methylation levels of 14 CpG sites of BDNF exon IX in patients with MDD and the healthy controls. (* *p* < 0.05)

**Figure 2 jcm-08-00568-f002:**
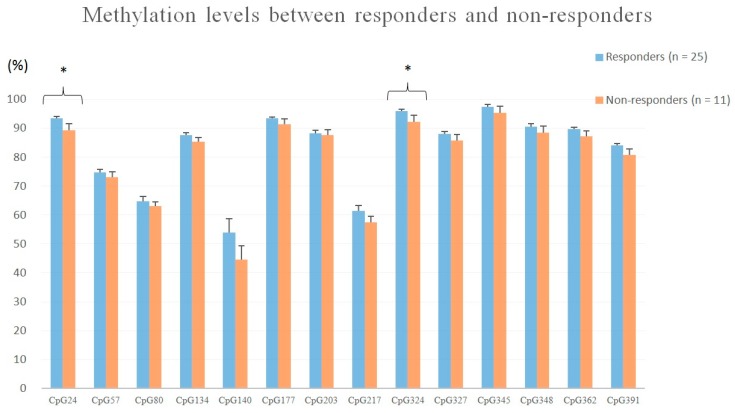
Methylation levels of 14 CpG sites of BDNF exon IX between drug responders and non-responders. (* *p* < 0.05)

**Figure 3 jcm-08-00568-f003:**
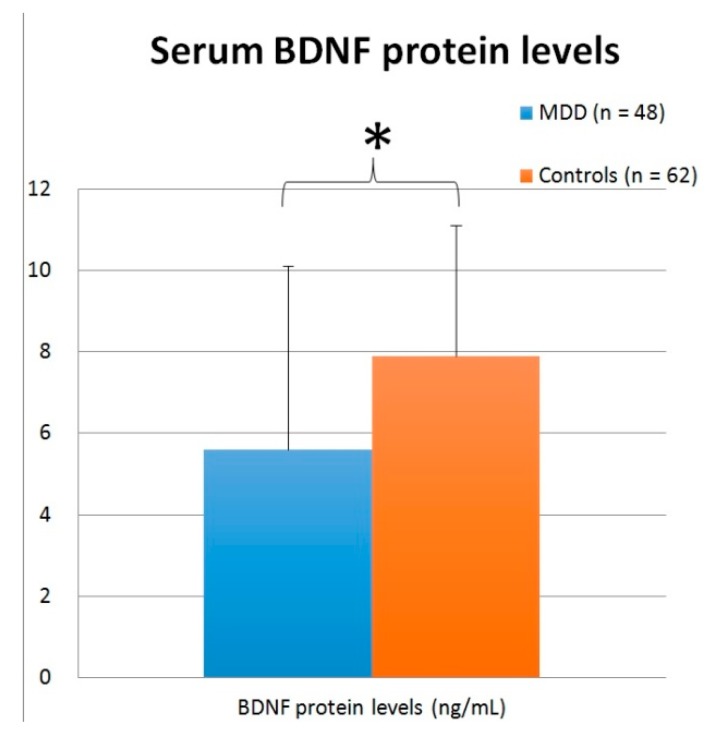
Serum BDNF protein levels (ng/mL) in patients with MDD and the healthy controls. (* *p* < 0.05)

**Figure 4 jcm-08-00568-f004:**
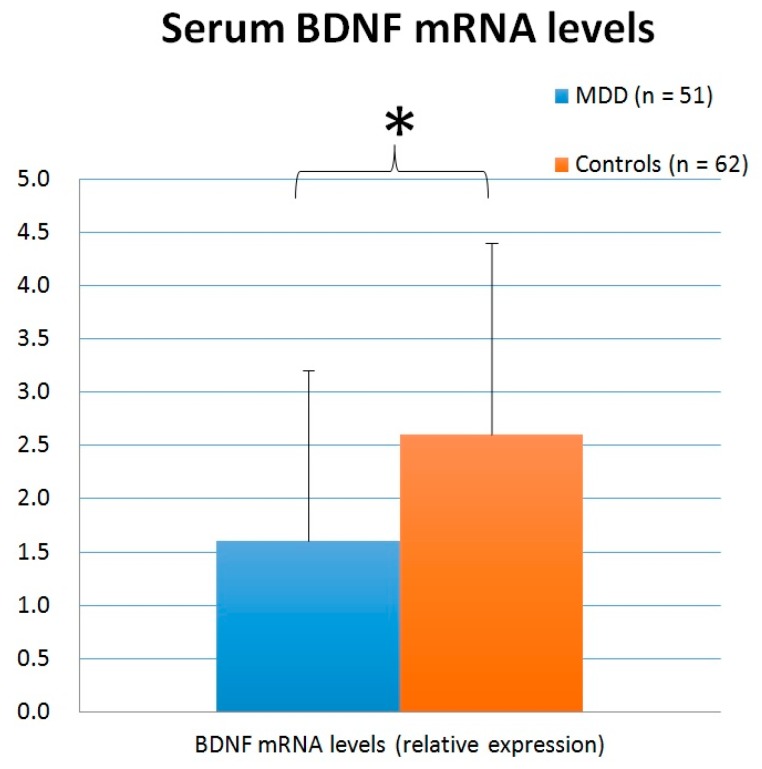
Serum BDNF mRNA levels (relative expression) in patients with MDD and the healthy controls. (* *p* < 0.05)

**Table 1 jcm-08-00568-t001:** Demographic data and serum BDNF mRNA levels at baseline of all participants.

Diagnostic Groups	Age	BMI (kg/m^2^)	Education (years)	Duration of Illness (years)	Hamilton-17 Item Rating Score	BDNF mRNA Levels
Healthy Controls (*n* = 62)	30.2 ± 5.6	22.4 ± 4.2	17.4 ± 1.8			2.6 ± 1.8
Men (*n* = 23)	27.2 ± 4.1	24.2 ± 4.3	18.0 ± 3.2			2.6 ± 1.4
Women (*n* = 39)	32.0 ± 5.7	21.4 ± 3.9	17.2 ± 1.0			2.6 ± 2.0
MDD (*n* = 51)	42.2 ± 11.8	22.5 ± 4.4	12.2 ± 3.3	5.0 ± 5.7	33.1 ± 4.9	1.6 ± 1.6
Men (*n* = 16)	41.3 ± 10.4	24.0 ± 5.2	12.5 ± 3.2	4.9 ± 5.7	34.4 ± 3.4	0.9 ± 0.5
Women (*n* = 35)	42.6 ± 12.5	21.9 ± 3.9	12.0 ± 3.4	5.0 ± 5.8	32.5 ± 5.4	1.8 ± 1.9

BDNF: brain-derived neurotrophic factor; BMI = body mass index; MDD: major depressive disorder

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
