# Peer review of "Abnormal Brain-Derived Neurotrophic Factor Exon IX Promoter Methylation, Protein, and mRNA Levels in Patients with Major Depressive Disorder"

_jcm, 2019, doi:10.3390/jcm8050568_

Reviewer 1 Report

Overall the findings in the manuscript meet the expectations for publication, however minor edits are necessary to finalize acceptance. 

Figure 1 is good and comprehensive, however the results throughout the paper are difficult to follow with this one figure.   If the figure were to be split into multiple focused smaller figures, categorized as per results topic, the manuscript would have a more logical flow. 

Editing Notes:
a. Minor typos throughout the manuscript. i.e line 198 promoter 

b. When inserting acronyms, keep consistency throughout the manuscript. i.e MDD

Table 1-some aspects can be graphed for better representation of the results than what is currently there. i.e BDNF levels

Conclusion can be slightly clearer, as to how various degrees of methylation and BDNF is effecting specific signalling mechanisms linked to MDD.
The clarity can be added in terms of what the authors postulate is happening downstream of DNA methylation. 

Author Response

Overall the findings in the manuscript meet the expectations for publication, however minor edits are necessary to finalize acceptance. 

 Figure 1 is good and comprehensive, however the results throughout the paper are difficult to follow with this one figure.   If the figure were to be split into multiple focused smaller figures, categorized as per results topic, the manuscript would have a more logical flow. 

 Ans: Thank you for your suggestions. We have added the * signs to indicate the CpG sites differentially expressed between patients and controls. That should make the figure much easier to follow.

 Editing Notes: 
a. Minor typos throughout the manuscript. i.e line 198 promoter 

 Ans: Thank you for pointing out the typo. It's been corrected.

 b. When inserting acronyms, keep consistency throughout the manuscript. i.e MDD

 Ans: We proofread the manuscript and made changes as suggested.

 Table 1-some aspects can be graphed for better representation of the results than what is currently there. i.e BDNF levels

 Ans: Thank you for your suggestions. We added two more figures comparing BDNF protein and mRNA levels.

 Conclusion can be slightly clearer, as to how various degrees of methylation and BDNF is effecting specific signalling mechanisms linked to MDD.
The clarity can be added in terms of what the authors postulate is happening downstream of DNA methylation. 

 Ans: Thank you for your suggestion. We have added the following sentences in the conclusion: " The mainstream theory is that increased BDNF promoter methylations would lead to impaired DNA expression, thus a decrease in protein production. However, different studies tend to utilize different approaches, such as the number of CpG sites surveyed (from 4 to 35 in our citations) and how their methylation levels were calculated (as individual levels vs. a total or global level). Those methodological differences make it difficult to draw an unifying conclusion to develop reliable diagnostic or outcome prediction tools, but as we gain more and more understandings in the interconnecting mechanisms of various pathways, we should have a better understanding of the underlying pathophysiology of psychiatric disorders."

Reviewer 2 Report

The aim of the present paper was to test methylation in BDNF DNA, protein and mRNA level in peripheral blood in control and MDD patients. Results are different in control and patients. Methylation increased in Patients responding to treatment with no variation in protein and mRNA level. While the general idea is interesting there are many criticism on both the way data have been presented and conclusions. It is difficult to understand the relevance of present data in clinical practice and if it is possible to use them to develop a diagnostic tool to evaluate the efficacy of treatments.

Table 1 is clear and easy to read, I wonder why the authors did not include also the data of protein level together to mRNA.  When these data are discussed (lines 178-179) values are mixed up (higher for patients).

Fig. 1 is the problem the authors claim a difference between control and patients and even more a change in patients responding to treatment. It could be wise to compare data (control patient-responding, patient-non responding) in a graph or a table.

The discussion is too long and not well organized and it is difficult to understand what is the message that the authors want to convey.

Author Response

The aim of the present paper was to test methylation in BDNF DNA, protein and mRNA level in peripheral blood in control and MDD patients. Results are different in control and patients. Methylation increased in Patients responding to treatment with no variation in protein and mRNA level. While the general idea is interesting there are many criticism on both the way data have been presented and conclusions. It is difficult to understand the relevance of present data in clinical practice and if it is possible to use them to develop a diagnostic tool to evaluate the efficacy of treatments.

 Ans: Thank you for your suggestion. We have added the following sentences in the conclusion: " The mainstream theory is that increased BDNF promoter methylations would lead to impaired DNA expression, thus a decrease in protein production. However, different studies tend to utilize different approaches, such as the number of CpG sites surveyed (from 4 to 35 in our citations) and how their methylation levels were calculated (as individual levels vs. a total or global level). Those methodological differences make it difficult to draw an unifying conclusion to develop reliable diagnostic or outcome prediction tools, but as we gain more and more understandings in the interconnecting mechanisms of various pathways, we should have a better understanding of the underlying pathophysiology of psychiatric disorders."

 Table 1 is clear and easy to read, I wonder why the authors did not include also the data of protein level together to mRNA.  When these data are discussed (lines 178-179) values are mixed up (higher for patients).

 Ans: Thank you for your correction! The mRNA numbers were indeed mixed up, so we corrected them. Only 48 patients had BDNF protein levels data, as opposed to 51 patients with mRNA data, so it is somewhat awkward to include BDNF protein levels in the table. We added two more figures to better present the BDNF protein and mRNA levels.

 Fig. 1 is the problem the authors claim a difference between control and patients and even more a change in patients responding to treatment. It could be wise to compare data (control patient-responding, patient-non responding) in a graph or a table.

 Ans: Thank you for your suggestions. We have added the * signs to indicate the CpG sites differentially expressed between patients and controls. That should make the figure much easier to follow.

 The discussion is too long and not well organized and it is difficult to understand what is the message that the authors want to convey.

 Ans: Thank you for your suggestion. We have revised the discussions with more transitional sentences as well as decreasing some jargons to make it easier to read.

Round  2

Reviewer 2 Report

papers have been improved but it is still missing the comparison between patients responding and not responding to treatment

Author Response

papers have been improved but it is still missing the comparison between patients responding and not responding to treatment

 Ans: Thank you for the reminder! We have added another figure (Figure 2) comparing responders and non-responders.